# Information Difference of Transfer Entropies between Head Motion and Eye Movement Indicates a Proxy of Driving

Runlin Zhang [1], Qing Xu [1,*], Shunbo Wang [1], Simon Parkinson [2] and Klaus Schoeffmann [3]

[1] College of Intelligence and Computing, Tianjin University, Tianjin 300072, China; runlin@tju.edu.cn (R.Z.); wang.shunbo@foxmail.com (S.W.)
[2] Department of Computer Science, University of Huddersfield, Huddersfield HD1 3DH, UK; s.parkinson@hud.ac.uk
[3] Institute of Information Technology, Klagenfurt University, 9020 Klagenfurt, Austria; klaus.schoeffmann@aau.at
* Correspondence: qingxu@tju.edu.cn

**Abstract:** Visual scanning is achieved via head motion and gaze movement for visual information acquisition and cognitive processing, which plays a critical role in undertaking common sensorimotor tasks such as driving. The coordination of the head and eyes is an important human behavior to make a key contribution to goal-directed visual scanning and sensorimotor driving. In this paper, we basically investigate the two most common patterns in eye–head coordination: "head motion earlier than eye movement" and "eye movement earlier than head motion". We utilize bidirectional transfer entropies between head motion and eye movements to determine the existence of these two eye–head coordination patterns. Furthermore, we propose a unidirectional information difference to assess which pattern predominates in head–eye coordination. Additionally, we have discovered a significant correlation between the normalized unidirectional information difference and driving performance. This result not only indicates the influence of eye–head coordination on driving behavior from a computational perspective but also validates the practical significance of our approach utilizing transfer entropy for quantifying eye–head coordination.

**Keywords:** head–eye coordination; unidirectional information difference; transfer entropy; behaviometrics

## 1. Introduction

Visual scanning performed under the effort of eye, head and torso is important for general human environment interactions [1,2]. The investigation of visual scanning provides a fundamental window into the nature of visual-cognitive processing while performing naturalistic sensorimotor tasks such as walking and driving [3,4]. The underpinning mechanism of visual scanning and visual-cognitive processing essentially includes the coordination of head and eyes in the procedure of performing sensorimotor tasks [5–9]. Therefore, head–eye coordination can be used as a valid means to study the internal mechanisms of visual scanning and visual cognitive processes.

Head–eye coordination primarily exhibits two patterns: "head motion earlier than the eye movement" and "eye movement earlier than the head motion" [7]. The "head motion earlier than the eye movement" is frequently observed in goal-directed, top-down and prepared tasks [5]. Conversely, the "eye movement earlier than the head motion" often occurs in stimulus-driven, bottom-up and spontaneous tasks [10]. Thus, when head and eye movements occur concurrently, the pattern of head–eye coordination reflects the level of preparedness for the gaze shift, which subsequently affects the performance of sensorimotor tasks. Therefore, this paper quantitatively measures the state of head–eye coordination from the perspective of its patterns, aiming to explore the relationship between head–eye coordination and driving performance.

We design a virtual reality driving task to obtain the head motion and eye movement data that we need to investigate. Firstly, driving is one of the most common sensorimotor tasks and a popular topic, and head–eye coordination is abundantly present in driving. Secondly, the driving task, as a whole, is executed as a "top-down" goal-directed activity [1,2]. During driving, "head motion earlier than eye movement" should dominate, which aids in achieving more significant results.

Eye movement and head motion data are observed as time series of eye rotation $\langle X_t \rangle$ and of head rotation $\langle Y_t \rangle$, respectively, labeled with a sequential time index $t = \cdots, 1, 2, \cdots$. In this paper, stochastic processes, usually used as natural representations for complex and real-world data [11], are introduced to model the time series data of eye movement and head motion, denoted by variables $X$ and $Y$, respectively.

Therefore, the behavior of head–eye coordination is reflected in the inter-relationship between $X$ and $Y$ [12]. For example, if the coordination of "head motion earlier than eye movement" exists, the past of head motion $Y_{t-1}$ helps predict the current observation of eye movement $X_t$. That is to say, the probabilistic predictivity of $X_t$ is added to $Y_{t-1}$. The transfer entropy from head motion to eye movement ($TE_{Y \rightarrow X}$) precisely measures this contribution [13], the same as transfer entropy from eye movement to head motion ($TE_{X \rightarrow Y}$).

Based on this, we use the transfer entropy between head movement and eye movement to measure head–eye coordination during driving [13]. Firstly, significant $TE_{Y \rightarrow X}$ can provide evidence for the presence of "head motion earlier than eye movement" coordination, while significant $TE_{X \rightarrow Y}$ can demonstrate the existence of "eye movement earlier than head motion" coordination. Secondly, according to the Wiener and Granger causality [14], the unidirectional information difference ($UID$) between $TE_{Y \rightarrow X}$ and $TE_{X \rightarrow Y}$ can determine whether the coordination between the head and eyes occurs from the head to the eyes or from the eyes to the head.

Notice that, although the research on the dynamics of the coordination of head and eyes in visual scanning attracted a lot of studies recently [5–7,9], there is no quantitative measure on this coordination. The tight connection between the information flow between head motion and eye movement and head–eye coordination leads us to believe that quantifying head–eye coordination based on transfer entropy is feasible. In this paper, we introduce the normalized unidirectional information difference ($NUID$), which preserves the relationship between unidirectional information difference and head–eye coordination, makes $TE_{Y \rightarrow X}$ and $TE_{Y \rightarrow X}$ into the same scale and improves the unidirectional information difference. We have found a significant correlation between driving performance and the normalized unidirectional information difference from head motion to eye movement. Our finding indicates that head–eye coordination during driving, with a quantification based on transfer entropy, is related to driving performance.

This paper is organized as follows. Firstly, related works are presented in Section 2. We then describe the proposed methodology for the new measures in Section 3. The experiment conducted is detailed in Section 4, followed by the results and discussion in Section 5. Finally, we present the conclusion and future works in Section 6.

## 2. Related Works

### 2.1. Transfer Entropy

Transfer entropy, basically as a measure of complexity, is a well-known way for quantifying the directional information flow between time series [13]. Transfer entropy is considered as a non-parametric and model-free version of the Wiener and Granger causality [14], being capable of handling complex and non-linear time series [11]. Given random variables $P$ and $Q$, transfer entropy from source $Q$ to target $P$ is defined as follows [11]:

$$\begin{aligned}
TE_{Q \rightarrow P}^{(l,k)} &= I(P_t : \boldsymbol{Q}_{t-1}^{(l)} | \boldsymbol{P}_{t-1}^{(k)}) \\
&= H(P_t | \boldsymbol{P}_{t-1}^{(k)}) - H(P_t | \boldsymbol{P}_{t-1}^{(k)}, \boldsymbol{Q}_{t-1}^{(l)}),
\end{aligned} \tag{1}$$

where $P_t$ and $Q_t$ are the observations of variables $P$ and $Q$ at time $t$, respectively, $\boldsymbol{P}_{t-1}^{(k)} = (P_{t-k}, \cdots, P_{t-1})$ and $\boldsymbol{Q}_{t-1}^{(l)} = (Q_{t-l}, \cdots, Q_{t-1})$ are the temporally ordered histories of target and source variables, respectively, and $H(\cdot|\cdot)$ and $I(\cdot : \cdot)$ represent, respectively, conditional entropy and mutual information. Here, $l$ and $k$ are the so-called history lengths of $\boldsymbol{Q}_{t-1}^{(l)}$ and of $\boldsymbol{P}_{t-1}^{(k)}$, respectively. Notice that the information flow from $Q$ to $P$ obtained via $TE_{Q \to P}^{(l,k)}$ tries to take out the influences of the past of $P$.

Transfer entropy is asymmetric. Because $H(P_t|\boldsymbol{P}_{t-1}^{(k)})$ is no smaller than $H(P_t|\boldsymbol{P}_{t-1}^{(k)}, \boldsymbol{Q}_{t-1}^{(l)})$, transfer entropy is non-negative. Considering that $H(P_t|\boldsymbol{P}_{t-1}^{(k)}, \boldsymbol{Q}_{t-1}^{(l)})$ and $H(P_t|\boldsymbol{P}_{t-1}^{(k)})$ are non-negative, $TE_{Q \to P}^{(l,k)}$ takes $H(P_t|\boldsymbol{P}_{t-1}^{(k)})$ as the maximum.

### 2.2. Coordination of Head and Eyes

Recently, many research studies have suggested the large popularity of the coordination of head and eyes in human activities [8,9,15,16], for example, in motor control [8,9,15,16].

The coordination of head and eyes always exists in our behavioral activities, particularly when a relatively large attentional shift is about to occur [1,2,17–19]; as a matter of fact, this coordination emerges as long as the eye movement is bigger than $15°$ [9,17]. Specifically, the coordination of head and eyes is necessary because eye movements could selectively allocate the available attentional resources to task relevant information and head motions could accommodate the limited field of view of the eyes [18,19]. That is, head motions and eye movements are synergistic, especially temporally, for visual scanning and visual-cognitive processing [6]. Basically, head motions are followed by eye movements (namely, the preparatory head motion earlier than the eye movement) during sensorimotor tasks, because the observer usually has prior and "top-down" knowledge, attaining attentional shift for goal-directed modulation [6–9,20].

Note that head–eye coordination involving head motions temporally preceding eye movements (rather than coordination with eye movements temporally preceding head motions) has been definitively accepted as the main coordination of head and eyes in goal-directed human activities [6–9] and principally contributes to goal-directed modulation during sensorimotor tasks [1,2,17–19]. In addition, the point here is that the directional coordination of head and eye movements itself does possess information about the performer's attentional and cognitive state, affecting task performance [7–9,21,22].

### 2.3. Complexity Measures for Visual Scanning

In this paper, the complexity measures based on information entropy, which have been used for the assessment of visual scanning efficiency, are introduced.

The entropy rate can be identified by multiplying the summation of inverse transition durations and the normalized entropy of fixation sequence together [23]. The entropy of fixation sequence (*EoFS*) is the *Shannon* entropy of the probability distribution of fixation sequences [24]. Gaze transition entropy (*GTE*) [25] is defined as a conditional entropy based on the probability transition between *Markov* states (namely, the areas of interest (*AOIs*)). Stationary gaze entropy (*SGE*) [25] gives the *Shannon* entropy based on an equilibrium distribution of *Markov* states. The latest technique called time-based gaze transition entropy (*TGTE*) [9], which uses time bins to realize the idea of *GTE*, is proposed for handling visual stimuli with dynamic changes.

## 3. The Proposed Methodology for New Measures

### 3.1. A Unidirectional Information Difference (UID)

As discussed in Section 1, head–eye coordination can be exploited as a measure of the unidirectional information difference. Following (1), transfer entropy from head motion $Y$ to eye movement $X$, $TE_{Y \to X}$, is defined as:

$$TE_{Y \to X} = H\left(X_t \Big| X_{t-1}\right) - H\left(X_t \Big| X_{t-1}, Y_{t-1}\right)$$
$$= \sum_{\substack{x_t, \\ x_{t-1}, y_{t-1}}} p\left(x_t, x_{t-1}, y_{t-1}\right) \log_2 \frac{p(x_t | x_{t-1}, y_{t-1})}{p(x_t | x_{t-1})}, \tag{2}$$

Note that in this paper, the history lengths of $X$ and $Y$ are both taken as 1, as usually performed in the literature [11]. Other possible options of the history length are outside of this paper's scope but will be considered in the near future. Here, $p(\cdot)$ and $p(\cdot|\cdot)$ denote the (conditional) probability distributions of gaze ($x_t$) and head ($y_t$) data. And similarly, $TE_{X \to Y}$, transfer entropy from eye movement to head motion, is given as follows:

$$TE_{X \to Y} = H\left(Y_t \Big| Y_{t-1}\right) - H\left(Y_t \Big| Y_{t-1}, X_{t-1}\right)$$
$$= \sum_{\substack{y_t, \\ y_{t-1}, x_{t-1}}} p\left(y_t, y_{t-1}, x_{t-1}\right) \log_2 \frac{p(y_t | y_{t-1}, x_{t-1})}{p(y_t | y_{t-1})}. \tag{3}$$

Notice that the more predictivity of current eyes ($X$) is added to the past of head ($Y$), the larger $TE_{Y \to X}$ is. Analogously, the more predictivity of the current head ($Y$) is added to the past of eyes ($X$), the larger $TE_{X \to Y}$ is. In this case, the unidirectional information difference from head motion $Y$ to eye movement $X$ can be defined as $TE_{Y \to X}$ minus $TE_{X \to Y}$:

$$UID_{Y \to X} = TE_{Y \to X} - TE_{X \to Y}. \tag{4}$$

It is easy to see that $UID_{Y \to X}$ is a methodology for identifying the Wiener and Granger causality [14]. When $UID_{Y \to X} > 0$, the causal relationship is from the head to the eyes, and the head–eye coordination presents as "head motion earlier than eye movement". When $UID_{Y \to X} < 0$, the causal relationship is from the eyes to the head, and the head–eye coordination represents "eye movement earlier than head motion". $UID_{Y \to X} = 0$ (practically $UID_{Y \to X}$ approaches zero) means that causality between the eyes and the head is not clear and that the head–eye coordination behaves ambiguously. In addition, the reason for using $TE_{Y \to X}$ minus $TE_{X \to Y}$ instead of $TE_{X \to Y}$ minus $TE_{Y \to X}$ is that we found $TE_{Y \to X}$ is statistically significant but $TE_{X \to Y}$ is not and the value of $TE_{Y \to X}$ is larger than $TE_{X \to Y}$ (see details in Section 5.2).

### 3.1.1. Significance Test

Measurement variance and estimation bias usually occur when obtaining transfer entropy, which is a common consideration [11]. Here, we take a hypothesis testing approach [26] to combat this problem.

The standard statistical technique of hypothesis testing [11,26,27], due to its popular use in handling time series data, is performed for determining whether there exists a valid $UID_{Y \to X}$ with a high confidence level. To accomplish this, the null hypothesis $H_0$ taken is that $UID_{Y \to X}$ is small enough, that is, it means that $X$ and $Y$ do not influence each other. And $H_1$ supports a causal-effect relationship between $X$ and $Y$, unidirectionally. To verify or reject $H_0$, surrogate time series $X_i^S$ and $Y_i^S$ ($i = 1, \cdots, N_S$) of the original $X$ and $Y$, respectively, are used. For surrogate generation, random shuffle, which is simple yet effective, is utilized, because in this paper, the history lengths of $X$ and $Y$ are both taken as 1, as usually used for the practical definition and computation of transfer entropy [11]. The unidirectional information difference from $Y_i^S$ to $X_i^S$, following (4), is obtained as follows:

$$UID_{Y_i^S \to X_i^S} = TE_{Y_i^S \to X_i^S} - TE_{X_i^S \to Y_i^S}. \tag{5}$$

The significance level of $UID_{Y \to X}$ is defined as:

$$\lambda_{Y \to X} = \frac{UID_{Y \to X} - \mu_{Y_i^S \to X_i^S}}{\sigma_{Y_i^S \to X_i^S}}, \tag{6}$$

where $\mu_{Y_i^S \to X_i^S}$ and $\sigma_{Y_i^S \to X_i^S}$ are the mean and standard deviation of $UID_{Y_i^S \to X_i^S}$ values, respectively. The probability of rejecting $H_0$ can be obtained based on Chebyshev's inequality, calculated as follows:

$$P(|UID_{Y \to X} - \mu_{Y_i^S \to X_i^S}| \geq k\sigma_{Y_i^S \to X_i^S}) \leq \frac{1}{k^2} = \alpha, \tag{7}$$

where $1 - \alpha$ is the confidence level of rejecting $H_0$ (and of accepting $H_1$) and parameter $k$ is any positive real number. The number of surrogates, which is related to the confidence level, is obtained as:

$$N_S = \frac{2}{\alpha} - 1 \tag{8}$$

for a two-sided test.

In this paper, the parameter $k$ used in (7) is taken as 6, resulting in a confidence level of 97.3%, and this is a high requirement satisfied in practice [27]. That is, if the significance level is bigger than 6 ($\lambda_{Y \to X} > 6$), then, equivalently, with a confidence level of more than 97.3%, there exists a unidirectional head–eye information flow from head motion to eye movement (note this technique is called $6 - Sigma$ [26]; some other techniques based on a $p - value$ approach to statistical significance testing [28] could be attempted in the future). In fact, according to statistical test theory [27], it is important to know that a minimum confidence level, acceptable in practice, is 95.0% (here, the corresponding significance level is 4.47). Notice that the significance and confidence levels play the same role in hypothesis testing.

The $UID_{Y \to X}$ and $UID_{Y_i^S \to X_i^S}$ computations, highlighted in red and blue boxes, respectively, are illustrated in Figure 1. The example values of $UID_{Y \to X}$ and $UID_{Y_i^S \to X_i^S}$ based on the gaze and head data of participant 5 in Trial 3 in our psychophysical studies are also presented (here, $UID_{Y \to X}^\star$ and $\lambda_{Y \to X}^\star$ are especially used for emphasis; see all the results relevant to the unidirectional information difference in Section 5.3). Clearly, a big difference between $UID_{Y \to X}^\star = 0.068$ (with a very high confidence level of 99.3% and a very large significance level $\lambda^\star$ of 12.53) and $UID_{Y_i^S \to X_i^S}$ ($\mu_{Y_i^S \to X_i^S} = 0.001$, $\sigma_{Y_i^S \to X_i^S} = 0.005$, $i = 1, \cdots, N_S$) exists. For the driving activity of participant 5 in Trial 3, there appears a significant unidirectional head–eye information difference from head motion to eye movement in goal-directed sensorimotor tasks.

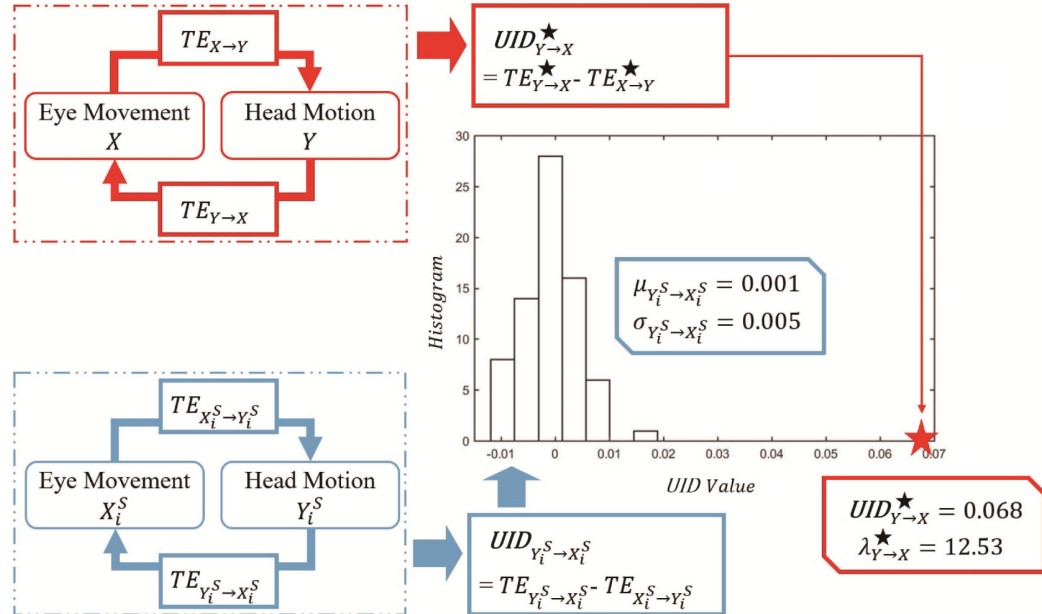

**Figure 1.** An illustration scheme for the computation of unidirectional information difference $UID_{Y \to X}$.

It is noticed that the significance test for the computation of the unidirectional information difference described here is standard and general enough to be employed as well for checking the statistical significance of the transfer entropy, as shown in Section 5.2.

### 3.2. A Normalized Unidirectional Information Difference (NUID)

In a goal-directed driving scenario, head–eye coordination corresponds to the state of visual scanning and visual-cognitive processing (correspondingly, the attentional states of drivers) [6–9], and meanwhile, this state signifies the performance of sensorimotor tasks [21,22]. As discussed in Section 3.1, the unidirectional information difference from head motion to eye movement in effect gives a quantitative estimation of the head–eye coordination. Therefore, we hypothesize that the unidirectional information difference should work well as a proxy of the driving performance. This hypothesis will be verified by using the correlation analysis technique, which is a classic and popular tool for investigating the relationship between variables [29].

Because a proxy indicator of driving performance actually contributes to an objective and quantitative score, for the sake of comparing performances, we propose a normalized unidirectional information difference from head motion to eye movement, $NUID_{Y \to X}$, for being quantitatively compatible with driving performance, as follows:

$$NUID_{Y \to X} = NTE_{Y \to X} - NTE_{X \to Y}, \tag{9}$$

where

$$NTE_{Y \to X} = \frac{TE_{Y \to X} - \mu_{Y^S \to X}}{H(X_t | X_{t-1})} \tag{10}$$

is a kind of normalized transfer entropy, whose definition is effective and popularly used [11]. Here, $\mu_{Y^S \to X}$ is the mean of the transfer entropies $TE_{Y_i^S \to X_i}$ $(i = 1, \cdots, N_S)$ from surrogate head motion to original eye movement, and the conditional entropy $H(X_t | X_{t-1})$ denotes the maximum of $TE_{Y \to X}$. $NTE_{X \to Y}$ can be obtained similarly:

$$NTE_{X \to Y} = \frac{TE_{X \to Y} - \mu_{X^S \to Y}}{H(Y_t | Y_{t-1})}. \tag{11}$$

Note other normalization methods for transfer entropy and for the unidirectional information difference could be performed in future work [30,31].

By normalization, both $NTE_{Y \to X}$ and $NTE_{X \to Y}$ are constrained to the range between $-0.5$ and $0.5$. Consequently, the range of $NUID_{Y \to X}$ is from $-1$ to $1$. $NUID_{Y \to X}$ differs from $UID_{Y \to X}$ when they take the zero value. For $NUID_{Y \to X}$, the zero value no longer signifies the primary direction for assessing causality or specific types of head–eye coordination. In the meantime, $NUID_{Y \to X}$ retains an important property. That is, the larger $NUID_{Y \to X}$ is, the more it indicates a tendency toward "head motion earlier than eye movement", while a smaller $NUID_{Y \to X}$ suggests a tendency toward "eye movement earlier than head motion". It is this property that leads us to choose $NUID_{Y \to X}$ to calculate the correlation with driving performance.

## 4. Experiment

### 4.1. Virtual Reality Environment and Task

Driving, which is commonly considered as a goal-directed activity [17,18], is taken as the sensorimotor task in our psychophysical experiments. Due to its repeatable usability, high safety and good performance, the (head-worn) virtual reality technique has become a popular paradigm to study gaze shifts in sensorimotor tasks [19,32–34]. Therefore, our study is performed based on head-worn virtual reality.

In this paper, the virtual environment for the psychophysical studies utilizes a four-lane, two-way, suburban road consisting of straight sections, curves (4 left bends and 4 right bends with mean radii of curvature of 30 m) and 4 intersections, with common trees and

buildings. In order to focus on the study of goal-directed activity in sensorimotor driving and on quantitatively investigating the *specific head–eye coordination* (with head motions temporally preceding eye movements), irrelevant visual distractors such as the sudden appearance of a running animal, which have been considered as ignored in the performing of goal-directed tasks [35] (and also this topic relevant to irrelevant visual distractors has been understood well in the research area [36]), are not included.

In our study, a single driving task, which is to smoothly maintain the driving speed at 40 km/h, is used. The inverse of the average acceleration during driving is taken as the indicator of driving performance, as popularly performed in the literature [37]. That is, the larger the average acceleration is, the worse the driving performance becomes, and *vice versa*.

Example illustrations of the virtual environment and of performing a driving task are presented in Figure 2.

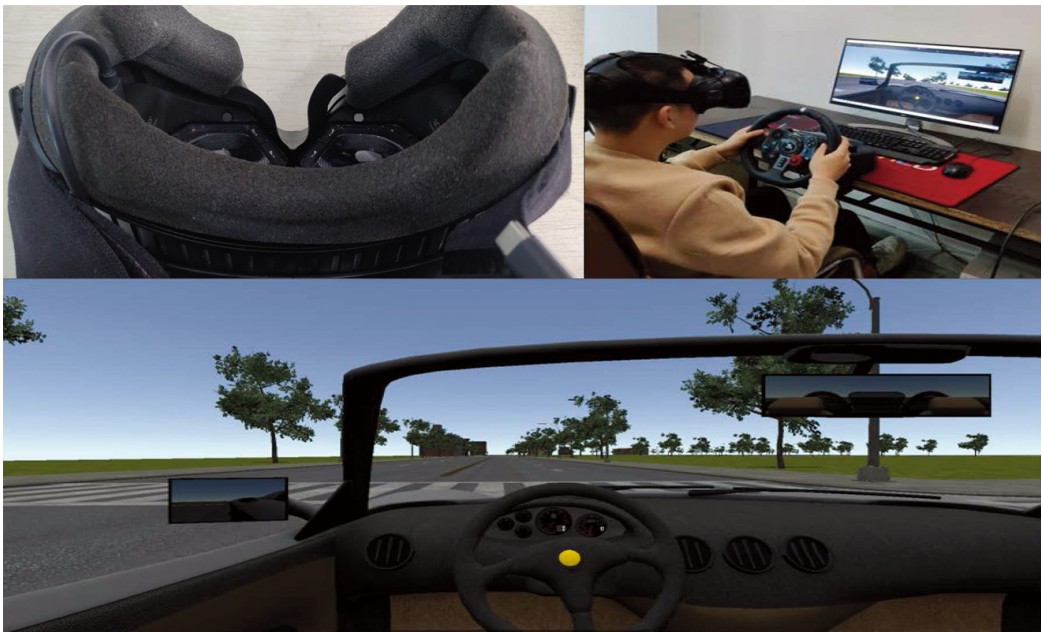

**Figure 2.** In the virtual environment (**bottom**), with an *HTC Vive* headset and a *7INVENSUN Instrument aGlass DKII* eye tracker (**top left**), a participant is performing the driving task (**top right**).

### 4.2. Apparatus

The psychophysical experiments in this paper are conducted in a virtual reality environment through the display via an *HTC Vive* headset [38]. And there is a *7INVENSUN Instrument aGlass DKII* eye-tracking piece of equipment [39] embedded in the headset. An illustration of the headset with the embedded eye tracker is given in Figure 2. Eye rotation and head motion (head rotation) data are recorded at a frequency of 90 Hz via the eye-tracking equipment (gaze position's accuracy is $0.5°$) and via the headset, respectively, both being captured as pitch and yaw (as usually conducted in the relevant field [7]). Virtual driving is performed based on a *Logitech G29* steering wheel [40]. A desktop monitor is utilized to display the captured data and driving activities of participants in the procedure of the experiment.

### 4.3. Participants

Twelve people participated in the psychophysical study. Each participant took part in four independent test sessions to have a large enough sample size for our study (see details in Section 4.4). These participants, with normal color vision and normal/corrected-to-normal visual acuity, were recruited from students at one of the authors' universities (7 male, 5 female; ages $22.9 \pm 1.95$). All of the participants held their driver licenses for no

less than one and a half years. None of the participants had any adverse reactions to the virtual environment utilized in this study. All participants provided written consent and were compensated with payment. This study was approved by the Ethics Committee of one of the authors' universities under the title "Eye tracking based Quantitative Behavior Analysis in Virtual Driving".

*4.4. Procedure*

Each participant finished four test sessions, with an interval of one week between every two consecutive tests, based on the same task requirements and driving routes. In this study, a test session is represented as a trial. In total, there were $12 * 4 = 48$ valid trials accomplished in the psychophysical experiments. Although this number of trials satisfies the large-sample condition in classical statistics [41], in the near future, a larger sample size could be utilized for making our proposed measures have more possible contributions to practical behaviometrics applications.

Before each test, the purpose and procedure of psychological studies were introduced to the participants. For the sake of high-quality data recordings, (a) all participants completed a 9-point calibration procedure prior to the experiments; (b) the headset was adjusted and fastened to participants' heads; (c) sight and eye cameras were adjusted to prevent hair and eyelashes from obscuring and (d) the seat was adjusted to a comfortable position in front of the steering wheel.

For each test session, first of all, conducting a 3-min period of familiarization was introduced. Then, for a 3-min driving session, participants were instructed to comply with driving rules: driving smoothly at a speed of 40 km/h and following the formulated routes (trying to stay close to the center line).

## 5. Results and Discussion

*5.1. Temporal Sequences of Head Motion and Eye Movement Data*

Example data for head motion and eye movement are plotted as a function of time, shown in Figure 3. As usually performed in the study of the coordination of head motion and eye movement [7,9], the data of eye and head rotations in yaw are utilized in this paper. It is obvious that head motion and eye movement always exist during driving. Furthermore, the synchronized registration of the local extreme values of head motion and eye movement data indicates, to a certain extent, an overall correspondence between two kinds of data, clearly showing that the coordination of head and eyes does exist. We introduce an evaluation measure for the amount of coordination of head and eyes (*CoordAmount*), inspired by the widely used measure *PSNR* in the field of signal processing [42], as follows:

$$CoordAmount = 10 \times \log_{10}\left(\frac{ScaleFactor^2}{Diff}\right), \tag{12}$$

where

$$Diff = \frac{1}{num}\sum_{t=1}^{num}(y_t - x_t)^2 \tag{13}$$

is the mean square difference (*Euclidean* distance) between gaze and head rotation data (*num* is the number of time units (*t*) considered). *ScaleFactor* = 360 is the maximal absolute difference between any two gaze and head data pair. *CoordAmount* quantifies the quality of matching two kinds of rotation data streams according to data values and shows the synergy of both rotation streams, providing a normalized measurement of the amount of synergistic coordination of head and eyes. The greater amount of coordination exists, the higher the *CoordAmount* becomes, and *vice versa*. The *CoordAmount* values, which are 31.45 dB and 33.09 dB for participants 1 (fourth trial) and 5 (third trial), respectively (in Figure 3), are relatively high, and this verifies the existence of the coordination of head and eyes. Note these two amounts of coordination for the two trials are close.

However, in fact, due to the lack of time sequence in its definition, *CoordAmount* can only be used to determine the presence of head–eye coordination but cannot ascertain

whether the coordination occurs with head motion preceding eye movement or vice versa. For example, the results from *CoordAmount* indicate that participants 1 (fourth trial) and 5 (third trial) exhibited coordination between head and eye movements during the driving process. However, it is only through a detailed analysis that we can determine whether the head moves first or the eyes move first. In Figure 3, we have highlighted two specific instances of head motion earlier than eye movement behaviors using boxes. In the box of the upper row, the head yaw starts to consistently increase from its local minimum earlier than the eye yaw, and in the box of the lower row, the head yaw starts to consistently decrease from its local maximum earlier than the eye yaw. In addition, the corresponding performance values are relatively diverse, 0.34 and 0.42 for the two participants, respectively. The latter is 1.24 times as large as the former. Therefore, relying solely on *CoordAmount* to determine the presence of head–eye coordination is insufficient. We need to ascertain whether the head moves before the eyes or vice versa, which one dominates the entire driving process and what their relationship is to driving performance. These are all questions worthy of our attention.

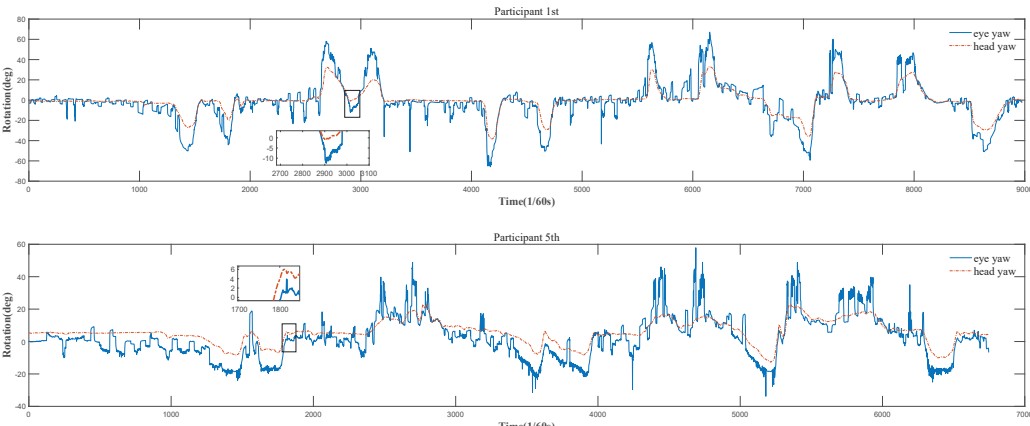

**Figure 3.** Examples of head and eye rotation data from the fourth and third trials of two participants, 1 and 5, upper and lower rows, respectively.

*5.2. Transfer Entropies between Head Motion and Eye Movement*

We first determine whether "head motion earlier than eye movement" and "eye movement earlier than head motion" exist. Both "head motion earlier than eye movement" and "eye movement earlier than head motion" have a significant impact on analyzing whether the previous moment's head motion (eye movement) has a notable influence on the current moment's eye movement (head motion). Therefore, we opt to use transfer entropy to characterize this process, as discussed in Section 3.1. The larger $TE_{Y \rightarrow X}$ is, the more predictivity of current X adds to the past of Y. Therefore, if $TE_{Y \rightarrow X}$ is statistically significant, it indicates that during the driving process, head motion has conveyed a substantial amount of information to eye movements, providing evidence for the existence of the head–eye coordination "head motion earlier than eye movement". The same applies to $TE_{X \rightarrow Y}$.

All the values of transfer entropies are listed in Table 1. A significant difference between two transfer entropies $TE_{Y \rightarrow X}$ and $TE_{X \rightarrow Y}$ is revealed via one-way analysis of variance ($ANOVA$) ($F(1, 94) = 80.25, p < 0.05$), as illustrated in Figure 4. The transfer entropy in the direction from head motion to eye movement, $TE_{Y \rightarrow X}$, is much bigger than that in the reverse direction, $TE_{X \rightarrow Y}$, with the averages of the former and latter $3.8 \times 10^{-2}$ and $1.9 \times 10^{-2}$, respectively. That is, $TE_{Y \rightarrow X}$ is twice as big as $TE_{X \rightarrow Y}$ for the experimentation data in this paper. Further, statistical significance testing, which is completely similar to what has been described in Section 3.1.1, is used for checking the statistical confidence levels of $TE_{Y \rightarrow X}$ and $TE_{X \rightarrow Y}$, entirely separately. It is observed that the significance and confidence levels for $TE_{Y \rightarrow X}$ are 4.49 and 95.0%, respectively. In contrast, the two corresponding values for $TE_{X \rightarrow Y}$ are 1.46 and 53.4%, respectively.

This means that $TE_{Y \to X}$ and $TE_{X \to Y}$ are statistically acceptable and unacceptable at 5% significance level, respectively. As previously mentioned, in goal-directed tasks, "head motion earlier than eye movement" is the primary pattern of head–eye coordination [5]. The result, where $TE_{Y \to X}$ is significant and $TE_{X \to Y}$ is not, validates our idea of using transfer entropy to measure the existence of head–eye coordination patterns.

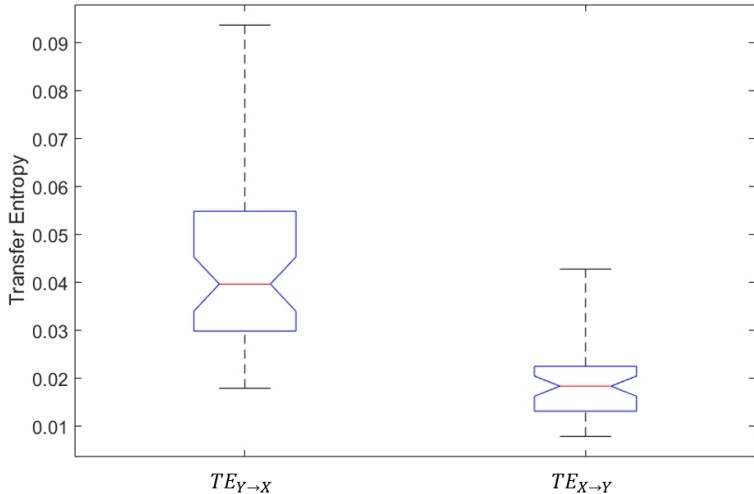

**Figure 4.** Transfer entropies between head motion and eye movement.

**Table 1.** Values of transfer entropies $TE_{X \to Y}$ and $TE_{Y \to X}$ ($\times 10^{-2}$).

| Participant | Trial 1 | | Trial 2 | | Trial 3 | | Trial 4 | |
|---|---|---|---|---|---|---|---|---|
| | $X \to Y$ | $Y \to X$ | $X \to Y$ | $Y \to X$ | $X \to Y$ | $Y \to X$ | $X \to Y$ | $Y \to X$ |
| 1 | 2.19 | 2.98 | 1.99 | 2.77 | 1.68 | 1.85 | 1.75 | 2.99 |
| 2 | 1.82 | 3.03 | 2.60 | 6.93 | 1.64 | 3.52 | 2.00 | 4.76 |
| 3 | 2.25 | 3.68 | 1.97 | 2.58 | 2.42 | 2.72 | 2.66 | 4.48 |
| 4 | 0.79 | 1.97 | 1.25 | 2.23 | 1.31 | 3.16 | 1.17 | 3.86 |
| 5 | 1.82 | 4.96 | 1.13 | 5.87 | 0.86 | 7.66 | 1.59 | 3.84 |
| 6 | 1.83 | 5.38 | 4.28 | 8.71 | 2.06 | 5.09 | 1.99 | 3.96 |
| 7 | 1.45 | 4.36 | 2.99 | 6.41 | 2.26 | 6.60 | 1.09 | 4.74 |
| 8 | 1.39 | 2.31 | 1.49 | 3.51 | 3.00 | 6.86 | 2.31 | 3.88 |
| 9 | 1.76 | 5.09 | 2.17 | 6.50 | 1.85 | 5.60 | 2.16 | 3.79 |
| 10 | 1.12 | 1.80 | 1.23 | 4.30 | 1.08 | 2.99 | 1.86 | 4.95 |
| 11 | 2.27 | 5.02 | 1.22 | 2.43 | 2.09 | 3.98 | 1.10 | 2.80 |
| 12 | 1.32 | 3.58 | 2.34 | 7.87 | 2.81 | 8.99 | 2.36 | 9.37 |

Furthermore, the lack of statistical significance in $TE_{X \to Y}$ does not necessarily imply the absence of eye movement followed by head motion throughout the entire driving process. Rather, it signifies that the influence of head movements on eye movements during driving is minimal. In such cases, we conclude that there is no significant head–eye coordination with "eye movement earlier than head motion" during the driving process.

### 5.3. The Unidirectional Information Difference between Head Motion and Eye Movement

This section primarily investigates the primary pattern of head–eye coordination during the driving process, whether it is "head motion earlier than eye movement" or "eye movement earlier than head motion". We employ a commonly used approach in Wiener and Granger causality analysis, which involves calculating the difference in information transfer between the two directions. We observe that $TE_{Y \to X}$ is greater than $TE_{X \to Y}$ and statistically significant at a 5% significance level, whereas $TE_{X \to Y}$ is not statistically significant. Therefore, we conclude that "head motion earlier than eye movement" predominates during the driving process, while "eye movement earlier than head motion" is

attributed to data variability. Therefore, we opt for $TE_{Y \to X}$ as the minuend and $TE_{X \to Y}$ as the subtrahend when calculating the information difference. This choice ensures a positive information difference and enhances the interpretability of its underlying meaning.

The unidirectional head–eye information difference $UID_{Y \to X}$ results (as provided in Table 2) are obtained with high significance levels ($\lambda_{Y \to X}$), which are presented in Table 3. Almost all the $\lambda_{Y \to X}$ values are larger than 6, that is, the corresponding confidence levels are more than 97.3%. There are only two exceptional evaluations of $\lambda_{Y \to X}$, 5.7 and 5.5, marked with boxes (Table 3), that are slightly lower than 6. Even here, the corresponding confidence levels are 96.9% and 96.7%, respectively, and this is acceptable in statistics for practical use [43]. The strict positive $UID_{Y \to X}$ (Table 2) reveals that there indeed exists a unidirectional information difference from head motion to eye movement (with high confidence), in the procedure of performing goal-directed sensorimotor tasks.

**Table 2.** Values of unidirectional information difference $UID_{Y \to X}$ ($\times 10^{-2}$).

| Participant | Trial 1 | Trial 2 | Trial 3 | Trial 4 |
|:---:|:---:|:---:|:---:|:---:|
| 1 | 0.79 | 0.78 | 0.17 | 1.24 |
| 2 | 1.21 | 4.33 | 1.88 | 2.76 |
| 3 | 1.43 | 0.61 | 0.30 | 1.82 |
| 4 | 1.18 | 0.98 | 1.85 | 2.70 |
| 5 | 3.14 | 4.73 | 6.80 | 2.25 |
| 6 | 3.55 | 4.43 | 3.02 | 1.97 |
| 7 | 2.91 | 3.42 | 4.33 | 3.65 |
| 8 | 0.91 | 2.02 | 3.86 | 1.57 |
| 9 | 3.33 | 4.33 | 3.75 | 1.62 |
| 10 | 0.68 | 3.06 | 1.91 | 3.09 |
| 11 | 2.76 | 1.21 | 1.89 | 1.70 |
| 12 | 2.26 | 5.53 | 6.18 | 7.01 |

**Table 3.** Significance levels $\lambda_{Y \to X}$ for $UID_{Y \to X}$.

| Participant | Trial 1 | Trial 2 | Trial 3 | Trial 4 |
|:---:|:---:|:---:|:---:|:---:|
| 1 | 8.89 | 9.18 | 15.25 | 10.23 |
| 2 | 17.33 | 18.08 | 7.83 | 5.68 |
| 3 | 17.01 | 5.48 | 11.20 | 11.64 |
| 4 | 11.94 | 15.03 | 12.25 | 16.67 |
| 5 | 16.02 | 12.18 | 12.53 | 8.02 |
| 6 | 15.86 | 20.18 | 21.13 | 17.15 |
| 7 | 17.28 | 19.36 | 26.34 | 22.89 |
| 8 | 7.86 | 13.92 | 13.58 | 10.74 |
| 9 | 16.40 | 17.84 | 17.37 | 13.94 |
| 10 | 14.32 | 16.86 | 13.14 | 14.47 |
| 11 | 19.35 | 12.71 | 18.87 | 19.43 |
| 12 | 14.67 | 12.52 | 16.69 | 15.06 |

*5.4. The Normalized Unidirectional Information Difference between Head Motion and Eye Movement*

Now, we aim to quantitatively characterize the relationship between head–eye coordination and driving performance. We utilize the inverse of the average acceleration (denoted by $1/AvgAcc$) as a measure of driving performance. However, the correlation between $UID_{Y \to X}$ and driving performance was not statistically significant. Therefore, we improved $UID_{Y \to X}$ by adopting the normalization to obtain the normalized unidirectional information difference ($NUID_{Y \to X}$). Although $NUID_{Y \to X}$ alters the value range and the meaning of its zero value, it still indicates an important property for practice. That is, a higher $NUID_{Y \to X}$ points out a stronger tendency toward "head motion earlier than eye movements" and vice versa.

The results of the normalized head–eye unidirectional information difference ($NUID_{Y \to X}$) and the corresponding driving performance (the inverse of the average acceleration, denoted by $1/AvgAcc$) are listed in Table 4. As a concrete instance, depicted in

Figure 3 together with the corresponding descriptions in Section 5.1, the two very different $NUID_{Y \to X}$ values obtained by participants 1 and 5 (in the fourth and third trials) are $-0.18 \times 10^{-2}$ and $8.07 \times 10^{-2}$, respectively. The large difference between these two values of $NUID_{Y \to X}$ corresponds closely to the big difference between the two coordination patterns of head and eyes of these two participants and, meanwhile, contrasts sharply with the closeness of the two corresponding *CoordAmount* values. Importantly, this clearly reveals that the proposed $NUID_{Y \to X}$, which represents the degree of head–eye coordination pattern, is largely related to driving activity and performance. More importantly, $NUID_{Y \to X}$ even enhances discriminating to differentiate the distinct driving activities of the two participants under consideration (correspondingly, the two relatively diverse values of driving performances are 0.34 and 0.42, respectively, with the latter 1.24 times as big as the former). In fact, a significant correlation ($p < 0.05$) between the new normalized information difference and driving performance, based on all the head and gaze data in 48 trials, is obtained via three correlation analyses [29], with a Pearson linear correlation coefficient (*PLCC*), Kendall rank order correlation coefficient (*KROCC*) and Spearman rank order correlation coefficient (*SROCC*) of 0.32, 0.27 and 0.41, respectively (Table 5). These correlation coefficient values definitely indicate a statistically significant relationship between our proposal of normalized information difference and driving performance, as popularly recognized in the literature [44]. By contrast, the measurements using the compared techniques (Table 5) cannot show an acceptable association with the performance of virtual driving ($p > 0.05$).

**Table 4.** Values of normalized unidirectional information difference $NUID_{Y \to X}$ ($\times 10^{-2}$) and driving performance ($1/AvgAcc$).

| Participant | Trial 1 | | Trial 2 | | Trial 3 | | Trial 4 | |
|---|---|---|---|---|---|---|---|---|
| | $NUID_{Y \to X}$ ($\times 10^{-2}$) | $1/AvgAcc$ ($s^2/m$) | $NUID_{Y \to X}$ ($\times 10^{-2}$) | $1/AvgAcc$ ($s^2/m$) | $NUID_{Y \to X}$ ($\times 10^{-2}$) | $1/AvgAcc$ ($s^2/m$) | $NUID_{Y \to X}$ ($\times 10^{-2}$) | $1/AvgAcc$ ($s^2/m$) |
| 1 | $-1.75$ | 0.37 | $-1.13$ | 0.43 | $-0.35$ | 0.42 | $-0.18$ | 0.34 |
| 2 | 1.23 | 0.71 | 2.77 | 0.64 | 3.11 | 0.69 | 5.05 | 0.89 |
| 3 | 1.66 | 0.45 | 1.65 | 0.41 | $-0.80$ | 0.42 | $-0.28$ | 0.40 |
| 4 | 4.85 | 0.72 | 1.50 | 0.56 | 4.45 | 0.56 | 4.84 | 0.52 |
| 5 | 5.95 | 0.44 | 6.02 | 0.48 | 8.07 | 0.42 | 4.47 | 0.55 |
| 6 | 4.18 | 0.25 | $-1.42$ | 0.29 | 3.00 | 0.30 | 0.88 | 0.30 |
| 7 | 6.28 | 0.56 | 0.60 | 0.52 | 5.17 | 0.64 | 8.20 | 0.57 |
| 8 | $-2.54$ | 0.35 | 2.15 | 0.46 | 2.58 | 0.40 | 0.91 | 0.40 |
| 9 | 3.40 | 0.62 | 1.25 | 0.44 | $-0.99$ | 0.40 | $-1.04$ | 0.47 |
| 10 | 3.21 | 0.44 | 7.97 | 0.51 | 8.45 | 0.36 | 5.93 | 0.50 |
| 11 | 4.76 | 0.45 | 2.41 | 0.42 | 4.12 | 0.44 | 3.15 | 0.38 |
| 12 | 2.39 | 0.64 | 2.69 | 0.59 | 4.35 | 0.55 | 2.93 | 0.71 |

The statistically significant positive correlation between $NUID_{Y \to X}$ and driving performance may be due to the fact that, as the degree of the "head motion earlier than eye movement" increases, the driver's preparation for the gaze shift becomes more adequate, thus leading to better driving performance. This indicates that our experimental design is effective. $NUID_{Y \to X}$, by measuring the patterns of head–eye coordination, has established a correlation with driving performance. The mathematical essence of all the transfer entropy-relevant formulas in this paper is well suited for assessing and quantifying head–eye coordination. Prior to our research, no work had been able to demonstrate a significant correlation between driving performance and the transfer entropy-based measure of head–eye coordination. As a comparison, we also calculated other eye movement indicators mentioned in Section 2.3 and methods commonly used in signal processing to analyze the similarity between two signals, PSNR and SSIM [45]. We analyzed their relationship with driving performance, as shown in Table 5. Among all methods, only $NUID_{Y \to X}$ showed a significant effect on driving performance. Our studies have effectively translated

the abstract concept of head–eye coordination into an objective quantity and provided meaningful insight into its influence on driving. Furthermore, we believe our methodology offers a new perspective for digitizing similar abstract concepts.

**Table 5.** Correlation analysis between measures and driving performance.

| Methods | PLCC, *p*-Value | KROCC, *p*-Value | SROCC, *p*-Value |
|---|---|---|---|
| $NUID_{Y \rightarrow X}$ | **0.32**, $p < 0.05$ | **0.27**, $p < 0.05$ | **0.41**, $p < 0.05$ |
| *TGTE* | 0.19, $p > 0.05$ | 0.19, $p > 0.05$ | 0.26, $p > 0.05$ |
| *GTE* | 0.07, $p > 0.05$ | 0, $p > 0.05$ | −0.01, $p > 0.05$ |
| *SGE* | −0.07 $p > 0.05$ | −0.09, $p > 0.05$ | −0.15, $p > 0.05$ |
| *EoFS* | 0.01, $p > 0.05$ | 0, $p > 0.05$ | −0.03, $p > 0.05$ |
| *Entropy rate* | −0.06, $p > 0.05$ | −0.01, $p > 0.05$ | −0.02, $p > 0.05$ |
| *Fixation rate* | −0.24, $p > 0.05$ | −0.17, $p > 0.05$ | −0.24, $p > 0.05$ |
| *Saccade amplitude* | 0.25, $p > 0.05$ | 0.11, $p > 0.05$ | 0.09, $p > 0.05$ |
| *PSNR* | 0.08, $p > 0.05$ | 0.11, $p > 0.05$ | 0.14, $p > 0.05$ |
| *SSIM* | −0.17, $p > 0.05$ | −0.13, $p > 0.05$ | −0.21, $p > 0.05$ |

Bold indicates the indicators proposed in this paper.

## 6. Conclusions and Future Works

In this paper, we designed a "top-down" goal-directed driving experiment based on virtual reality to collect head and eye movement data from drivers. We treated head motion data and eye movement data as two stochastic processes and calculated transfer entropy from head to eye and from eye to head to determine the presence of head–eye coordination in terms of "head motion earlier than eye movement" and "eye movement earlier than head motion" . We discovered a significant existence of the head–eye coordination "head motion earlier than eye movement" among drivers during driving, while there was no clear evidence of "eye movement earlier than head motion" coordination. By calculating unidirectional information differences, we established that head–eye coordination predominates during driving. Without compromising the ability of $NUID_{Y \rightarrow X}$ to measure head–eye coordination patterns, we optimized unidirectional information differences, yielding normalized unidirectional information differences. Notably, we found a significant correlation between normalized unidirectional information differences and driving performance. This discovery validates two key points: firstly, head–eye coordination during the driving process does impact a driver's performance, and secondly, our approach of quantifying this abstract concept of head–eye coordination using transfer entropy is both feasible and meaningful in practice.

In the future, transfer entropy, unidirectional information differences and its normalized version can be applied to a broader range of abstract concepts, quantifying and validating their practical significance. During the resampling process, particularly in the resampling of multivariate time series, maintaining the auto-correlation of the time series could be utilized to analyze the correlation between head–eye coordination [46] and also to measure head–eye coordination. Furthermore, as mentioned in our paper, head–eye coordination is not the sole factor influencing driving performance. Beyond head–eye coordination, it is essential to identify additional elements that impact driving, allowing for a more precise modeling of driver behavior.

**Author Contributions:** Conceptualization, R.Z. and Q.X.; validation, R.Z.; formal analysis, R.Z.; investigation, R.Z. and S.W.; resources, Q.X.; data curation, R.Z.; writing—original draft preparation, R.Z.; writing—review and editing, Q.X., S.W., S.P. and K.S.; supervision, Q.X.; project administration, Q.X.; funding acquisition, Q.X. All authors have read and agreed to the published version of the manuscript.

**Funding:** This research was funded by the Natural Science Foundation of China under Grant No. 61471261 and No. 61771335 and was funded by The National Key Research and Development Program of China (Grant No. 2020YFC1807904 and No. 2020YFC1807905).

**Data Availability Statement:** The data presented in this study are openly available in https://github.com/zhangrlll/unidirectional-causality accessed on 15 December 2023.

**Conflicts of Interest:** The funders had no role in the design of the study; in the collection, analyses or interpretation of data; in the writing of the manuscript or in the decision to publish the results.

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
