# Peer review of "Information Difference of Transfer Entropies between Head Motion and Eye Movement Indicates a Proxy of Driving"

_entropy, doi:10.3390/e26010003_

Round 1

Reviewer 1 Report

Comments and Suggestions for Authors

This is an interesting paper that uses transfer entropies to conclude 'head motion earlier than eye movement' compared to 'eye movement earlier than head motion' regarding directionality of the head-eye coordination while driving.

Following comments need clarification:

1) For the surrogate time-series used in the calculation of significance level (page-5), authors mention that random-shuffle was used. Does it preserve the auto-correlation structure of the original time series. If not, why is that not relevant? Some explanation will be illuminating. It looks like similar strategies (based on random shuffle) have been employed in some papers mentioned in the references (correct?) But, I am puzzled why preserving the autocorrelation structure (as in in time-series bootstrap methods) is not important. See, for example, papers by D.N. Politis (UC-San Diego), especially Jentsch and Politis 2015. 

2) Notation-wise: shouldn't there be super-scripts-S as in UID_{Y^{S} --> X^{S}} in e.q. 6 since this is calculated under the null, H0?

3) The authors have to be careful in reporting results where the null-hypothesis was not rejected. E.g., in line 359 of. p.10, it should be something like, "acceptable and unacceptable at 5% significance level".

4) It may be a good idea to also comment upon the role of sample size in these results, especially in the tests where the null hypotheses were not rejected.

5) Finally, it is well-known that the 'time-spacing' can affect the Granger-Causality effects. Does any of that apply in this case as well?

Reviewer 2 Report

Comments and Suggestions for Authors The authors have investigated common patterns in eye-head coordination using transfer entropy. They have further established that the information theoretic measure transfer entropy is highly correlated to driving performance.  Following are some of my questions:   1. Why have the authors used transfer entropies and not any other measures? What is the rationale behind this?   2. I am not sure about the significance of this work. The authors have not provided any comparison with alternative methods. This is important because, apparently, the paper has no theoretical contributions.   3. Some of the claims made in this paper are very subjective and the authors should provide more justification about those claims. For example, this claim --> it is crucial to determine whether “head motion 337 earlier than eye movement” and “eye movement earlier than head motion” exists. Comments on the Quality of English Language

Minor edits are required.

Round 2

Reviewer 1 Report

Comments and Suggestions for Authors

With the revisions/modifications, I think the paper is acceptable for publication.

Reviewer 2 Report

Comments and Suggestions for Authors

I have no further comments.